# Prestrain-Enabled Stretchable and Conductive Aerogel Fibers

**DOI:** 10.3390/polym17212936

**Published:** 2025-11-01

**Authors:** Hao Yin, Jian Zhou

**Affiliations:** 1Naval University of Engineering, Wuhan 430033, China; 2School of Materials Science and Engineering, Key Laboratory for Polymeric Composite and Functional Materials of Ministry of Education, Guangzhou Key Laboratory of Flexible Electronic Materials and Wearable Devices, Laboratory of Advanced Electronics and Fiber Materials, Sun Yat-sen University, Guangzhou 510275, China

**Keywords:** stretchability, aerogel, fiber, wearable textile, conductivity

## Abstract

Aerogels combine ultralow density with high surface area, yet their brittle, open networks preclude tensile deformation and hinder integration into wearable electronics. Here we introduce a prestrain-enabled coaxial architecture that converts a brittle conductive aerogel into a highly stretchable fiber. A porous thermoplastic elastomer (TPE) hollow sheath is wet-spun using a sacrificial lignin template to ensure solvent exchange and robust encapsulation. Conductive polymer-based precursor dispersions are infused into prestretched TPE tubes, frozen, and lyophilized; releasing the prestretch then programs a buckled aerogel core that unfolds during elongation without catastrophic fracture. The resulting TPE-wrapped aerogel fibers exhibit reversible elongation up to 250% while retaining electrical function. At low strains (<60%), resistance changes are small and stable (ΔR/R_0_ < 0.04); at larger strains the response remains monotonic and fully recoverable, enabling broad-range sensing. The mechanism is captured by a strain-dependent percolation model in which elastic decompression, contact sliding, and controlled fragmentation/reconnection of the aerogel network govern the signal. This generalizable strategy decouples elasticity from conductivity, establishing a scalable route to ultralight, encapsulated, and skin-compatible aerogel fibers for smart textiles and deformable electronics.

## 1. Introduction

Aerogels are ultralight, highly porous solids (porosity > 99%) with exceptional surface area, enabling thermal insulation, liquid absorption, energy storage, and mechanical sensing [1,2,3]. Many aerogels tolerate compressive loading—sometimes with resilience in air or water [4,5]—but their open, brittle networks and thin cell struts render them intrinsically vulnerable under tension. This brittleness complicates integration with deformable systems and limits use in stretchable insulators and wearable electronics, where tensile compliance and robust encapsulation are mandatory [6,7].

Recent efforts have begun to address tensile deformation but remain constrained by either limited stretchability or a lack of packaging. Joint-welded CNT aerogels introduce fixed nodes and mobile segments to enhance mechanical stability, yet the tensile strain before failure remains <25% and the networks collapse at higher strains [8]. Three-dimensional printing routes—e.g., Ag-nanowire aerogels designed with anegative Poisson’s ratio [9] and graphene/CNT carbon aerogels with hierarchical buckling [10]—demonstrate recoverable tensile strains up to 20% and 200%, respectively. However, these aerogels are typically exposed to air without soft, compliant encapsulation, making them unsuitable for skin contact and vulnerable to environmental damage [11,12]. Thus, creating stretchable and encapsulated aerogels remains an unmet need. Our prior coaxial wet-spinning of cellulose nanofibril (CNF) aerogels within a cellulose acetate (CA)-rich sheath produced continuous, strong fibers in which the porous sheath facilitated solvent exchange and protected the core; however, tensile stretchability was limited (~6%) [13]. We subsequently showed that wrapping rigid conductors with a thermoplastic elastomer (TPE) sheath via coaxial wet spinning imparts tensile compliance while providing a porous conduit that assists solvent removal and microstructural control in the core [14,15]. These observations suggest that an elastic, porous sheath could simultaneously (i) enable efficient solvent exchange for aerogel formation and (ii) maintain mechanical integrity under large tensile strains [16,17,18].

Here, we report a prestrain-enabled coaxial route to TPE-wrapped conductive aerogel fibers that reconciles tensile compliance with robust encapsulation. Porous TPE hollow sheaths are wet-spun from a TPE/lignin dope into ethanol and delignified to yield a permeable channel; prestretched tubes are then infused with aqueous conductive dispersions, rapidly frozen, and lyophilized to form a continuous aerogel core. Releasing the prestretch programs a buckled core that unfolds during elongation while the elastic sheath preserves integrity. A strain-dependent percolation picture is adopted in which elastic decompression, interfacial sliding/debonding, and controlled fragmentation/reconnection govern resistance–strain coupling inside the encapsulated network. The same design rules can be generalized to diverse functional cores and integrated into deformable thermal, electrical, and multimodal fiber devices.

## 2. Experimental Details

### 2.1. Materials

Aqueous poly(3,4-ethylene dioxythiophene)/poly(styrene sulfonate) (1:2.5 PEDOT/PSS *w*/*w* ratio) dispersion (Clevios PH 1000) was purchased from HC Starck, Inc., Newton, MA, USA, Polystyrene-blockpolyisoprene-block-polystyrene (TPE) (styrene, 22 wt%), Lignin (alkali, Mw ∼10,000), and dichloromethane (DCM) were purchased from Sigma Aldrich. Ethanol was purchased from Fisher Chemical. Cellulose nanofibrils (CNFs) were derived from rice straw via combined TEMPO-mediated oxidation and mechanical blending, as reported previously [8]. Briefly, 1.0 g of rice straw cellulose was oxidized in an aqueous solution containing 0.016 g of TEMPO, 0.1 g of NaBr and 5 mmol NaClO at pH 10.0. After centrifugation and dialysis, the TEMPO-oxidized cellulose was blended (Vitamix 5200, Vita-Mix Corporation, Olmsted Falls, OH, USA) at 37,000 rpm for 30 min, centrifuged (5000 rpm, 15 min) to collect the supernatant to be concentrated using a rotary evaporator (Buchi Rotavapor R-114, Flawil, Switzerland) to 0.7 wt% and stored at 4 °C for preparation of stretchable aerogel fibers.

### 2.2. Preparation of Conductive Dispersions

A 0.7 wt% PEDOT/PSS dispersion was prepared by adding 2.0 g of distilled water into 5.5 g of aq. PEDOT/PSS at 1.1 wt%, and vigorously stirring for 30 min to form a homogeneous mixture. To prepare 0.7 wt% PEDOT/PSS/CNF dispersion, equal mass (5.5 g) of aq. PEDOT/PSS was added to aq. CNF, both at 0.7 wt% concentration, and vigorously stirring for 30 min to form a homogeneous mixture.

### 2.3. Preparation of Hollow TPE Fibers

TPE pellets (1.5 g) were first vigorously stirred in DCM (6 g) for 1 h, and then lignin (2.25 g) passed through a 50 µm mesh sieve was added to prepare a 38.5 wt% concentration of 2:3 *w*/*w* TPE-lignin *w*/*w* mixture which was stirred for 2 h. Each of the core (ethanol) and shell solutions (TPE/lignin in DCM) was loaded into 10 mL syringes and spun through respective inner (21 G) and outer (15 G) stainless steel needles at rates of 600 µL min^−1^ and 200 µL min^−1^, respectively, using Fusion 200 syringe pumps (Chemyx Inc., Stafford, TX, USA). The fibers were continuously spun into an ethanol bath at ambient temperature. The as-spun wet fibers were then soaked in a water bath for 60 min to remove lignin. Finally, the fibers were resinized with water and immersed in ethanol, and stored for the preparation of aerogel fibers.

### 2.4. Preparation of TPE-Wrapped Conductive Aerogel Fibers

First, one end of a 14 cm or 6 cm long TPE hollow fiber was fixed on a metallic frame, and a 27 G needle was placed inside the other end of the fiber. The side of the fiber with the needle was stretched to 100% or 500% strain and fixed on the metallic frame. The as-prepared PEDOT/PSS/CNF aqueous dispersion at 0.7 wt% was loaded into a 10 mL syringe and was injected into the stretched TPE hollow fibers using a Fusion 200 syringe pump (Chemyx Inc.) at 300 µL min^−1^. The aq. PEDOT/PSS/CNF in TPE hollow fibers were then frozen (in Liquid nitrogen, 5 min) and lyophilized (−50 °C, 1 d, Free Zone 1.0 L Benchtop Freeze Dry System, Labconco, Kansas City, MO, USA) to yield the TPE-wrapped conductive aerogel fibers. These freeze-dried fibers after relaxation were vapor annealed by methanol at 55° for 3 h to enhance their electrical conductivities via secondary doping of PEDOT/PSS, in which methanol redistributes/extracts PSS and drives a benzoid-to-quinoid conformational shift that planarizes PEDOT and strengthens π–π stacking [15]. To ensure strict comparability in the plots, the datasets shown in the main text are from 4 fibers drawn from one representative batch.

### 2.5. Characterization

Scanning electron microscopy (SEM) was performed on the cross-section, inner, and outer surface of the fibers using a Quattro S machine at high vacuum mode (Thermo Scientific, Waltham, MA, USA). All samples were sputtered with gold of 3–5 nm thickness before imaging. The diameter of the pores on the inner and outer surfaces was measured and averaged from 30 pores by ImageJ64. The cross-sectional dimension of the fibers was measured from SEM images, and their mass was measured by a balance with 0.1 mg resolution, to calculate the fiber density (*ρ_f_*). The porosity of the fibers (*P_f_*) was calculated as: *P_f_* = 1 − *ρ_f_*/*ρ_b_*, where *ρ_b_* is the bulk density of cellulose equals 1.3 g cm^−3^. The mechanical behavior of hollow TPE fiber, TPE-wrapped conductive aerogel fibers, was measured by a 5566 Instron universal testing machine at a constant 5% min^−1^ strain rate. Fiber samples 3 cm in length were coated with epoxy adhesive at each end to protect them from damage during clamping. The tensile strength, Young’s modulus, and elongation were collected from at least 10 samples for each formulation, and their average values and standard deviation are reported. The setup for electromechanical testing of the specimen with the loading/unloading of the sample is controlled by a 5566 Instron machine. The loading and unloading of the sample were controlled by a 5944 mechanical testing machine (Instron Corporation, Norwood, MA, USA). The fibers were first placed on a paper frame, and both ends of the 3 cm long fibers were connected with copper wires, which were first painted with conductive silver epoxy, then epoxy adhesive for electrode protection. The resistance change in the fibers was monitored by a 1252B digital multimeter.

## 3. Result and Discussion

To produce TPE-wrapped conductive aerogel fibers, a combination strategy of hollow fiber wet spinning, lignin removal, crack initiation, and aerogel formation process was applied (Figure 1). It illustrates a coaxial wet-spinning and post-treatment process to produce a hollow and elastic fiber, to be filled with aqueous conductive aerogel precursor under the pre-strained state. The spinning nozzle consists of coaxial inner and outer channels constructed with 21 and 15-gauge (G) needles, respectively. The hollow TPE/lignin composite fiber was first wet-spun using a mixed dope of TPE and lignin in dichloromethane (DCM) for the sheath and ethanol to fill the inner channel. High loading of lignin particles was added to TPE. It was used as a sacrificial template that would be removed to create a porous structure in TPE. The 38.5 wt% TPE/lignin (2:3 *w*/*w*) in DCM was spun into an ethanol coagulation bath where the ethanol inside and outside of the hollow fiber extracted DCM, causing simultaneous phase inversion in the sheath and enabling continuous spinning of a single TPE/lignin filament over 3 m long. The as-spun hollow fiber was immersed in water for 1 h to remove lignin and then transferred to an ethanol bath to be stored for making aerogel fibers. The ethanol-filled TPE hollow fibers were first air-dried at room temperature for 10 min. Then, one end of the TPE hollow fibers was fixed on a metallic frame, and a 27 G needle was placed inside the other end of the fiber. The side of the fiber with a needle was stretched to 100% strain and fixed on the metallic frame. We confirmed that the stretching initiates transverse crack formation on the TPE, which will be crucial for water sublimation in the following lyophilization process. The as-prepared PEDOT/PSS or PEDOT/PSS/CNF aqueous dispersion at 0.7 wt% was injected into the stretched TPE hollow fibers and was frozen in liquid nitrogen for 5 min, then freeze-dried in a lyophilizer (−50 °C for 2 d) to form a continuous conductive aerogel in the core.

The process begins with the wet-spinning of a hollow fiber, promoting lignin removal and the creation of micro-sized pores. The fiber is then subjected to stretching in a water bath, inducing transverse cracks. Subsequently, conductive ink is injected into the core of the stretched fiber, followed by a freezing and lyophilization step to form a conductive aerogel within the core.

To produce TPE-wrapped conductive aerogel fibers, a combination strategy of hollow fiber wet spinning, lignin removal, crack initiation, and aerogel formation process was applied (Figure 1). It illustrates a coaxial wet-spinning and post-treatment process to produce a hollow and elastic fiber, to be filled with aqueous conductive aerogel precursor under the pre-strained state. The spinning nozzle consists of coaxial inner and outer channels constructed with 21- and 15-gauge (G) needles, respectively. The hollow TPE/lignin composite fiber was first wet-spun using a mixed dope of TPE and lignin in dichloromethane (DCM) for the sheath and ethanol to fill the inner channel. High loading of lignin particles was added to TPE. It was used as a sacrificial template that would be removed to create a porous structure in TPE. The 38.5 wt% TPE/lignin (2:3 *w*/*w*) in DCM was spun into an ethanol coagulation bath where the ethanol inside and outside of the hollow fiber extracted DCM, causing simultaneous phase inversion in the sheath and enabling continuous spinning of a single TPE/lignin filament over 3 m long. The as-spun hollow fiber was immersed in water for 1 h to remove lignin and then transferred to an ethanol bath to be stored for making aerogel fibers. The ethanol-filled TPE hollow fibers were first air-dried at room temperature for 10 min. Then, one end of the TPE hollow fibers was fixed on a metallic frame, and a 27 G needle was placed inside the other end of the fiber. The side of the fiber with a needle was stretched to 100% strain and fixed on the metallic frame. We confirmed that the stretching initiates transverse crack formation on the TPE, which will be crucial for water sublimation in the following lyophilization process. The as-prepared PEDOT/PSS or PEDOT/PSS/CNF aqueous dispersion at 0.7 wt% was injected into the stretched TPE hollow fibers and was frozen in liquid nitrogen for 5 min, then freeze-dried in a lyophilizer (−50 °C for 2 d) to form a continuous conductive aerogel in the core.

Figure 2a,b illustrate the effect of 100% prestrain on the cross-sectional morphology of a hollow TPE fiber. The application of pre-strain results in the development of a porous aerogel structure within the fiber’s core. This transformation can be attributed to mechanical stresses that create micro-voids and fractures in the material, facilitating a transition from a dense to a porous state. Figure 2c,d provide a closer look at the porous structure, showing a network of interconnected voids and enhanced porosity. These features are critical as they suggest a potential increase in surface area and interconnectivity within the material, which can enhance properties such as ion transport, flexibility, and mechanical compliance under additional stress. Figure 2e,f display the internal structure of TPE fibers without pre-strain, characterized by a smooth, continuous film devoid of pores. The dense and homogeneous nature of these fibers indicates a significantly different mechanical and possibly functional behavior compared to their pre-strained counterparts. The absence of pores in the unstrained fibers suggests lower flexibility and potentially different electrical or thermal conductivity properties. This structural integrity, while robust, may limit the material’s effectiveness in applications requiring high flexibility and adaptability.

In situ SEM reveals a strain-programmable surface morphology for the TPE-encapsulated PEDOT aerogel fibers. At 0% strain (Figure 3a,b), the TPE sheath shows a micron-scale dimpled texture composed of closed surface micro-voids formed during removal of lignin particles of the sheath. The sheath remains continuous and fully covers the conductive core. Upon 100% tensile strain under SEM (Figure 3c,d), the surface texture elongates and transforms into a reversible microcrack/buckle pattern oriented predominantly perpendicular to the fiber axis. The cracks/buckles open under tension and close on unloading, preventing catastrophic fracture while accommodating large elongation. Mechanically, this behavior is consistent with prestrain-seeded surface buckling in a compliant TPE skin coupled to a porous, energy-dissipative core; electrically, the conductive network remains percolated because crack opening mainly modulates inter-fiber contact area rather than severing the pathway. This microcrack/buckle mechanism explains the high tensile compliance with recoverable conductivity reported here. We performed a simple gauge-length test on the TPE-encapsulated PEDOT aerogel fibers (Figure 3e). Two fibers with lengths of 7.4 and 7.8 cm were stretched from their initial length L_0_ to more than ε = 500% (i.e., L ≈ 6 L_0_) and subsequently unloaded. The samples recovered close to their original length with only a small residual change, visually confirming high stretchability and elastic recovery.

Figure 4a shows how the resistance change (ΔR/R_0_) and strain percentage evolve over an extended period during incremental cyclical loading. The fiber is subjected to increasing strain levels up to 250%, showing a corresponding increase in resistance. The staircase test (Figure 4a) establishes a quasi-linear increase in ΔR/R_0_ up to ~200% strain, beyond which fiber breakage initiates, causing a resistance increase. Upon release to 0% strain, resistance recovers fully, demonstrating the resilient nature of the aerogel-TPE composite structure. The progressive nature of this test simulates real-world applications where the material would undergo varying levels of strain. The change in ΔR/R_0_ with strain suggests that the material’s electrical properties are highly dependent on its deformation, likely due to changes in the physical structure of the aerogel, such as alignment or separation of conductive pathways. Figure 4b–d represent cyclical tests at constant maximum strains of 60%, 80%, and 100%, respectively. The uniformity in cyclical patterns indicates the material’s ability to recover its initial electrical properties after each deformation cycle. The consistent response and quick recovery in resistance values post-deformation suggest that the aerogel fiber possesses good elastic properties and durability. The increase in resistance at higher strains (especially visible over 200%) could indicate nearing or exceeding the elastic limit, where permanent structural changes begin to occur. Figure 3e presents the hysteresis loops at different strains, providing insight into the material’s energy dissipation and mechanical damping capabilities. These loops are essential for understanding the delay between applied mechanical strain and the electrical response. The broader loops at higher strains imply greater energy dissipation, which could be beneficial in applications requiring shock absorption. The non-linearity of these loops highlights the complex interplay between mechanical stress and electrical conductivity. Moreover, Figure 4f suggests the stability of resistance under a constant strain over a prolonged period of 10,000 s. The minimal variation in resistance change presents the material’s stability and endurance. The low fatigue under sustained strain is indicative of the material’s suitability for applications in environments requiring long-term reliability, such as structural health monitoring or wearable electronics. The mechanism lies in using prestrain to form a buckled aerogel core during strain release, transforming a brittle material into a highly stretchable one. Moreover, within this ambient range, the TPE sheath limits moisture uptake and shields the porous core. Neither phase is expected to undergo thermal transitions that disrupt conduction under the reported strains, so the signals remain stable over repeated loading in air. This strategy not only achieves elasticity for aerogels but also maintains electrical performance, particularly stable at low strains (less than 100%) for sensitive detection, while accommodating dramatic changes at high strains for broad-range sensing.

At rest (ε = 0%), the conductive aerogel network is compact with stable resistance. Under small strain (<60%), the network decompresses with little change in resistance. At moderate strain (60–150%), network sliding causes a gradual increase in resistance. Beyond 150% strain, fragmentation of pathways leads to a sharp rise in resistance. During release, reconnection and crack closing restore conductivity, though resistance remains slightly higher than the initial state due to irreversible microstructural changes.

Figure 5 provides a mechanistic bridge to the responses in Figure 4. In the low-strain regime (ε < 60%), elastic decompression preserves most conductive contacts, explaining the highly repeatable ΔR/R_0_–time traces and small hysteresis observed in the 60–100% cyclic tests (Figure 4b–d). Here, resistance changes are dominated by modest increases in contact/tunneling distances within a still-percolated network, giving sensitive yet stable signals suited for precise detection. As strain reaches 60–200%, sliding and partial debonding progressively deplete the effective contact area and enlarge tunneling gaps, producing the nonlinearity and loop width seen in the ΔR/R0 strain hysteresis (Figure 4e). Beyond 200% strain, fragmentation breaks percolation, which accounts for the step-like surges in the incremental “staircase” test (Figure 4a). Upon release, crack closure and pathway reconnection rapidly recover conductivity, while the residual offset (R_1_ > R_0_) aligns with the modest drift under long holds (Figure 4f) and reflects irreversible but limited microstructural rearrangement. As a result, the prestrain-enabled, TPE-confined aerogel core operates in two useful regimes: a low-to-moderate strain window with stable, repeatable signals for high-fidelity sensing, and a high-strain window that tolerates large deformation with a broad, monotonic electrical response, supporting both sensitive motion detection and wide-range strain monitoring. Prestained conductive aerogel fibers are demonstrated with reversible ≈ 250% elongation, enabled by a surface microcrack/buckle architecture that preserves percolation; the fibers show stable electromechanical response under ambient conditions with near-complete length recovery, positioning their strain window above typical filler-in-elastomer conductors.

## 4. Conclusions

We have demonstrated a simple route to tensile stretchability in aerogels by combining (i) a porous, protective TPE sheath formed by coaxial wet spinning and template removal with (ii) prestrain-programmed buckling of a conductive aerogel core produced by freeze-drying. This architecture delivers fibers that reversibly extend to 250%, provide stable and sensitive signals at low strain (<60%), and maintain a broad, monotonic response at high strain, recovering fully after large deformations. The electromechanical behavior arises from a controllable sequence of network decompression, contact sliding, and fragmentation/reconnection, which reconciles the traditionally incompatible requirements of ultralight porosity and tensile compliance. Continuous cyclic operations and long holds (10,000 s) with minimal drift underscore the durability of the encapsulated design. This work establishes prestrain-enabled, elastomer-confined aerogel fibers as a practical building block for next-generation flexible and wearable electronics.

## Figures and Tables

**Figure 1 polymers-17-02936-f001:**
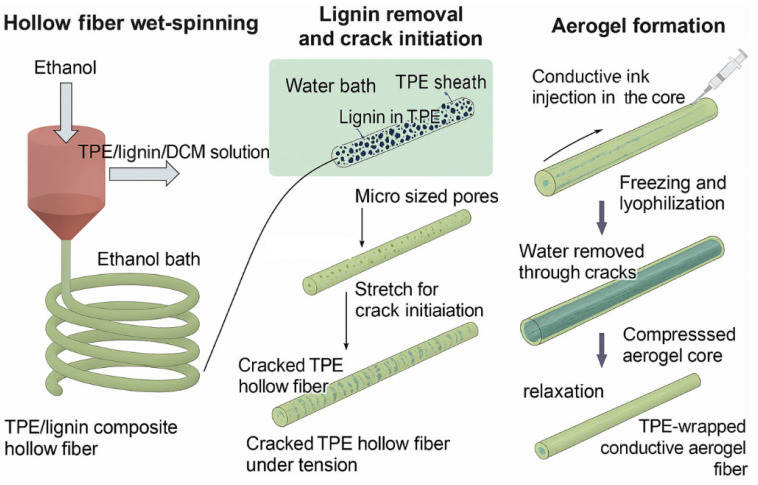
Fabrication process for the elastomer-wrapped aerogel fiber.

**Figure 2 polymers-17-02936-f002:**
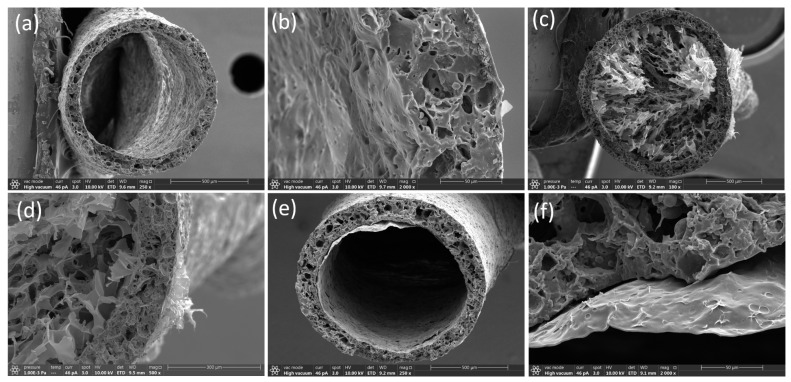
Morphology of TPE-encapsulated fibers with and without pre-strain. (**a**,**b**) Cross-section SEM image of a hollow TPE fiber following 100% pre-strain, presenting the formation of a porous structure. (**c**,**d**) Detailed SEM view of the porous aerogel structure in the core of a TPE/PEDOT fiber with 100% pre-strain, highlighting the enhanced porosity and network of voids created by the deformation process. (**e**,**f**) The SEM image of the film structure within the core of an unstrained TPE fiber emphasizes the smooth and continuous material distribution free of pores.

**Figure 3 polymers-17-02936-f003:**
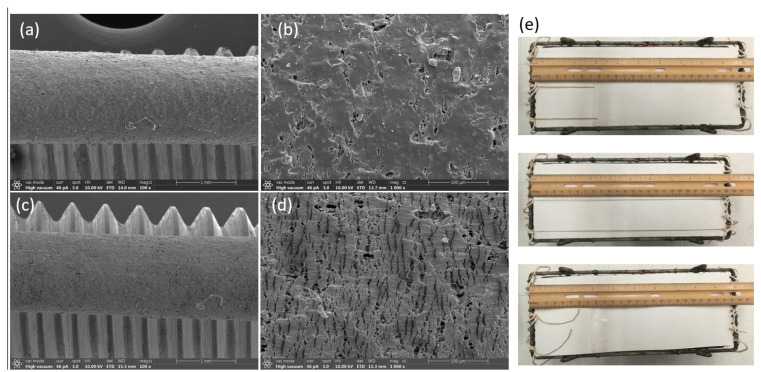
In situ SEM observation on the surface of porous TPE-encapsulated conductive fibers and demonstration of its large-strain stretchability and recoverability. (**a**,**b**) Surface SEM image of a TPE/PEDOT aerogel fiber (made from 100% pre-strain) under 0% strain, presenting the formation of a porous structure. (**c**,**d**) Surface SEM image of a TPE/PEDOT aerogel fiber (made from 100% pre-strain) under 100% strain, induces a reversible microcrack/buckle morphology that supports large tensile elongation without catastrophic fracture. (**e**) Photographic demonstration of large-strain stretchability and recovery of TPE/PEDOT aerogel fibers.

**Figure 4 polymers-17-02936-f004:**
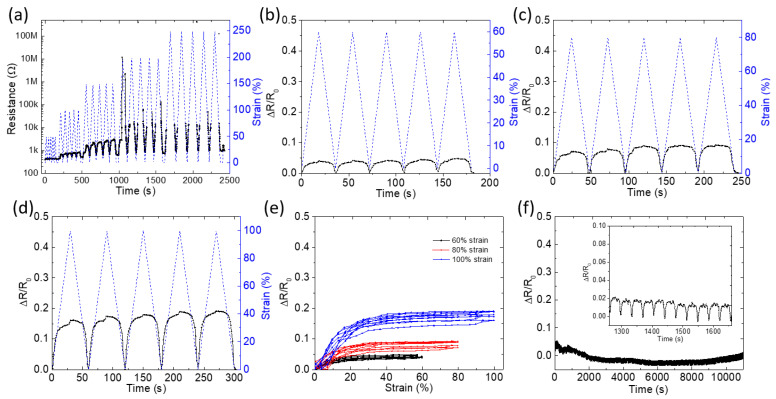
Electromechanical performance of TPE-wrapped aerogel fibers. (**a**) Resistance change (ΔR/R_0_) and strain during incremental stretching up to 250%. (**b**–**d**) Cyclic tests at 60%, 80%, and 100% strain show a stable resistance response. In (**a**–**d**), the blue dashed curves represent the applied strain, and the black solid curves represent the electrical resistance during cyclic loading. (**e**) Hysteresis loops of ΔR/R_0_ vs. strain highlight repeatable electrical behavior under strain. (**f**) Resistance stability under constant strain for 10,000 s indicates excellent durability.

**Figure 5 polymers-17-02936-f005:**
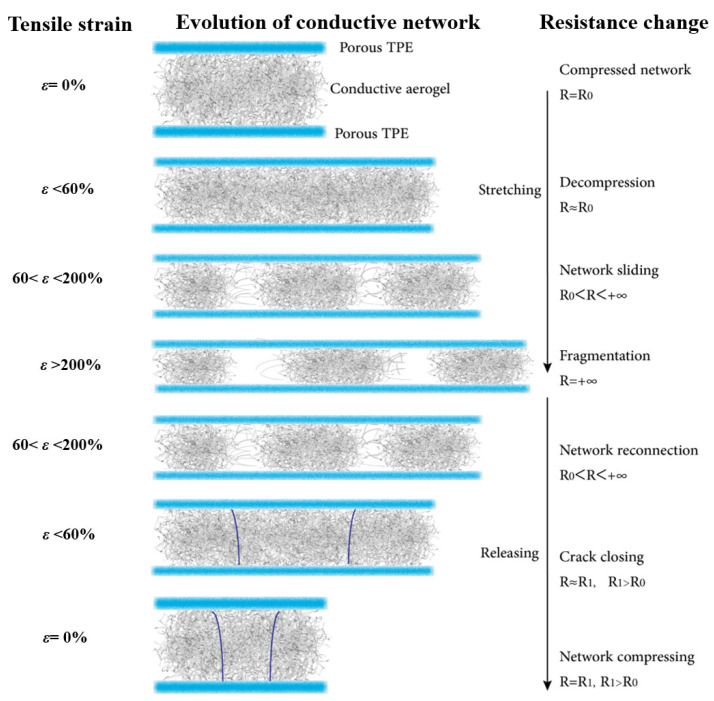
Mechanism of strain-dependent resistance change in TPE-wrapped aerogel fibers.

## Data Availability

The original contributions presented in this study are included in the article. Further inquiries can be directed to the corresponding authors.

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
