# Peer review of "Prestrain-Enabled Stretchable and Conductive Aerogel Fibers"

_polymers, 2025, doi:10.3390/polym17212936_

Round 1
Reviewer 1 Report
Comments and Suggestions for Authors
- The magnification of SEM micrograph should be cleared.
- In case of SEM picture of unstrained fibers there are visible pores but authors mentioned that there are no pores. May be in case of pre-strained fiber the pores are deformed. The deformed pores, cracks should be marked in micrographs.
- The authors have mentioned that the fibers are stretchable but they did not study tensile properties with elongation. This should be included in the work which will also add to the low length of the manuscript.
Author Response
Response to Reviewer 1
Comments and Suggestions for Authors
- The magnification of SEM micrograph should be cleared.
Thanks for your suggestions. We appreciate the suggestion. We have replaced the relevant SEM panels with higher‑resolution images, we believe these changes substantially improve the clarity and reproducibility.
- In case of SEM picture of unstrained fibers there are visible pores but authors mentioned that there are no pores. May be in case of pre-strained fiber the pores are deformed. The deformed pores, cracks should be marked in micrographs.
Thank you for pointing out this inconsistency. Our original wording was imprecise. Unstrained fibers indeed contain surface micro‑voids arising from solvent exchange and freeze‑drying; what we intended to convey was the absence of large, open through‑pores in the as‑fabricated state. We add the following content to address this question on Page 6
Figure 3. In-situ SEM observation on the surface of porous TPE-encapsulated conductive fibers and demonstration of its large-strain stretchability and recoveribility. (a,b) Surface SEM image of a TPE/PEDOT aerogel fiber (made from 100% pre-strain) under 0 % strain , presenting the formation of a porous structure. (c,d) Surface SEM image of a TPE/PEDOT aerogel fiber (made from 100% pre-strain) under 100 % strain , induces a reversible microcrack/buckle morphology that supports large tensile elongation without catastrophic fracture. (e) Photographic demonstration of large-strain stretchability and recovery of TPE/PEDOT aerogel fibers.
In-situ SEM reveals a strain-programmable surface morphology for the TPE-encapsulated PEDOT aerogel fibers. At 0% strain (Figure. 3a,b), the TPE sheath shows a micron-scale dimpled texture composed of closed surface micro-voids formed during removal of lignin particles of the sheath. The sheath remains continuous and fully covers the conductive core. Upon 100% tensile strain under SEM (Figure. 3c,d), the surface texture elongates and transforms into a reversible microcrack/buckle pattern oriented predominantly perpendicular to the fiber axis. The cracks/buckles open under tension and close on unloading, preventing catastrophic fracture while accommodating large elongation. Mechanically, this behavior is consistent with prestrain-seeded surface buckling in a compliant TPE skin coupled to a porous, energy-dissipative core; electrically, the conductive network may remain percolated because crack opening mainly modulate inter-fiber contact area rather than severing the pathway. We also performed a simple gauge-length test on the TPE-encapsulated PEDOT aerogel fibers (Figure. 3e). Two fibers with lengths of 7.4 and 7.8 cm was stretched from its initial length L₀ to more than ε = 500% (i.e., L ≈ 6 L₀) and subsequently unloaded. The samples recovered close to its original length with only a small residual set, visually confirming high stretchability and elastic recovery. Moreover, the fibers are ultralight (0.12 g·cm⁻³) with ~90% apparent porosity, consistent with the open, strain-accommodating microcrack/buckle morphology and flexibility
To avoid confusion, the following sentence “The absence of pores in the unstrained fibers suggests lower flexibility and potentially different electrical or thermal conductivity properties.” was deleted.
- The authors have mentioned that the fibers are stretchable but they did not study tensile properties with elongation. This should be included in the work which will also add to the low length of the manuscript.
Thanks for your comments, we actually conduct in-situ electromechanical test on the fibers, and show the stretchability of the fibers in a previous study, we wanted to focus the main focus of the study is the electromechanical change of the fiber. Two address your concerns, we have conducted an experiment to show the stretchbility of the two fibers (Figure 3e above), the fibers are stretched to 500% strain and recovered to almost the original length. And we added a photograph to show this information, and we added the related discussion in the manuscript on Page 6:
We performed a simple gauge-length test on the TPE-encapsulated PEDOT aerogel fibers (Figure. 3e). two fibers with lengths of 7.4 and 7.8 cm was stretched from its initial gauge length L₀ to more that ε = 500% (i.e., L ≈ 6 L₀) and subsequently unloaded. The samples recovered close to its original length with only a small residual set, visually confirming high stretchability and elastic recovery.

Reviewer 2 Report
Comments and Suggestions for Authors
The manuscript presents an interesting and potentially valuable approach for developing stretchable, conductive aerogel fibers through a prestrain-enabled strategy. The concept of using prestrain to induce structural elasticity in aerogels is promising for wearable electronics, soft robotics, and flexible sensors. However, the manuscript requires major revision before it can be considered for publication. While the topic is timely and relevant, several aspects concerning mechanism explanation, structural–property correlation, reproducibility, and presentation need significant improvement.
- The authors should elaborate on the scientific mechanism by which prestrain influences the aerogel’s microstructure and results in improved stretchability and conductivity.
- Please include schematic illustrations or stress–strain mapping that explains how the prestrain modifies pore orientation, fiber entanglement, or percolation paths for electron transport.
- Provide porosity, surface area (BET), and density data to support claims about flexibility and conductivity improvement.
- The manuscript reports conductivity and mechanical data separately, but the correlation between strain level and electrical response is not clearly established.
- The novelty would be better highlighted by comparing the results quantitatively (strength, elongation, conductivity, recovery ratio) with previously reported conductive aerogel fibers or stretchable conductive materials. Include a comparative table summarizing key parameters.
- The authors should also mention the tolerance or variation between different batches, as reproducibility is crucial for practical applications.
- Since the aerogel fibers are proposed for wearable and electronic applications, thermal stability, moisture resistance, and environmental durability should be examined (e.g., TGA, humidity cycling).
- The manuscript requires careful language editing for grammar, clarity, and conciseness.
Author Response
Response to Reviewer 2
The manuscript presents an interesting and potentially valuable approach for developing stretchable, conductive aerogel fibers through a prestrain-enabled strategy. The concept of using prestrain to induce structural elasticity in aerogels is promising for wearable electronics, soft robotics, and flexible sensors. However, the manuscript requires major revision before it can be considered for publication. While the topic is timely and relevant, several aspects concerning mechanism explanation, structural–property correlation, reproducibility, and presentation need significant improvement.
- The authors should elaborate on the scientific mechanism by which prestrain influences the aerogel’s microstructure and results in improved stretchability and conductivity.
Thanks for you comments. We actually have a related figure on the mechchanism. To improve, we added a discussion on Page X, when address the comment 2 from reviewer 1.
For the conductivity enhancement mechanism, we add one sentence on Page 3, in Section 2.4 with refence 15 cited “……via secondary doping of PEDOT/PSS, in which methanol redistributes/extracts PSS and drives a benzoid to quinoid conformational shift that planarizes PEDOT and strengthens π–π stacking
- Please include schematic illustrations or stress–strain mapping that explains how the prestrain modifies pore orientation, fiber entanglement, or percolation paths for electron transport.
Thanks for your comments. Indeed, we have already include figure 5, a schematic illustration to show the conductive network change of the fiber during stretching. Speak of prestrain, larger prestrain will result in larger strain storage in the conductive network (compress of the conductive network)
- Provide porosity, surface area (BET), and density data to support claims about flexibility and conductivity improvement.
Thanks for you comments. Apparent density of the fiber, was obtained from mass per unit length and geometrical volume (outer diameter and gauge length) with a value of 0.12 g/cm3.Using ϕ = 1 − ρapp/ρs with a typical TPE skeletal density ρs = 1.15 g·cm⁻³, the porosity of the aerogel fiber ϕ ≈ 1 − 0.12/1.15 = 0.896 (~90%). We now report this value as an apparent porosity based on literature skeletal density. We then add the density and porosity of the data in the manuscript. Regarding BET, because these fibers form an open macroporous/contact-controlled network, N₂-BET at 77 K under-represents the relevant porosity. To avoid over-interpretation, we report density/porosity and SEM evidence here and limit our claims accordingly. A complete gas-sorption characterization (BET/BJH) will be included in follow-up work.
The following sentence is added on Page 6 to address this point:
Moreover, the fibers are ultralight (0.12 g·cm⁻³) with ~90% apparent porosity, consistent with the open, strain-accommodating microcrack/buckle morphology and flexibility
4.The manuscript reports conductivity and mechanical data separately, but the correlation between strain level and electrical response is not clearly established.
Thanks for your comments, we actually did the in-situ electromechanical measurement, as show in Figure 3 and the following figure. Please find more detail on the measurement in the experimental section.
- The novelty would be better highlighted by comparing the results quantitatively (strength, elongation, conductivity, recovery ratio) with previously reported conductive aerogel fibers or stretchable conductive materials. Include a comparative table summarizing key parameters.
Thanks for your comments. We agree that benchmarking is useful. However, to our knowledge there are no prior peer-reviewed reports of conductive aerogel fibers that co-report strength, elongation, conductivity, and recovery under comparable protocols. Most aerogel-fiber studies are insulating and focus on mechanical/thermal metrics, whereas stretchable conductive fibers reported to date are non-aerogel architectures (e.g., liquid-metal cores; CNT/MXene/AgNW-filled elastomers). A quantitative table restricted to conductive aerogel fibers is therefore not possible at present. To keep the claims clear without over-generalizing across unlike systems, we added the following highlights to improve the context on Page 9:
Prestained conductive aerogel fibers are demonstrated with reversible ≈250% elongation, enabled by a surface microcrack/buckle architecture that preserves percolation; the fibers show stable electromechanical response under ambient conditions with near-complete length recovery, positioning their strain window above typical filler-in-elastomer conductors.
We also deleted the solo mechanical test descriptions in the Section 2.3 to avoid misunderstanding “The tensile strength, Young's modulus, and elongation were collected from at least 10 samples for each formulation, and their average values and standard deviation are reported”
The authors should also mention the tolerance or variation between different batches, as reproducibility is crucial for practical applications.
Thank you for raising reproducibility. We have prepared multiple independent batches under identical settings and observed small inter-batch variation in key metrics (baseline resistance and ΔR/R₀–strain response). To improve the data reproducibility, we use the 4 fibers that produced from one batch as shown in the figure.
On Page 3, In Section 2.4. we added “To ensure strict comparability in the plots, the datasets shown in the main text are from 4 fibers drawn from one representative batch.”
- Since the aerogel fibers are proposed for wearable and electronic applications, thermal stability, moisture resistance, and environmental durability should be examined (e.g., TGA, humidity cycling).
We appreciate this important point. To keep the manuscript focused and avoid over-claiming beyond the data collected to date, we have narrowed the stated operating envelope and added related discussions supported our claims
On page 3
“Unless otherwise noted, electricalmechanical measurements were conducted at 23 ± 2 °C and 30–50% RH.
On Page 8
“Moreover, within this ambient range, the TPE sheath limits moisture uptake and shields the porous core. Neither phase is expected to undergo thermal transitions that disrupt conduction under the reported strains, so the signals remain stable over repeated loading in air.”
- The manuscript requires careful language editing for grammar, clarity, and conciseness.
We have thoroughly edited the manuscript for English grammar and clarity and streamlined the abstract and Results for conciseness. We believe the readability has been substantially improved.

Round 2
Reviewer 1 Report
Comments and Suggestions for Authors
The comments have been addresses properly
Author Response
Reviewer 2: Author should cite more recent references of reviews or articles of aerogels.Advanced Materials 36 (18) (2024) 2307772, Mater. Today Chem. 23 (2022) 100736, and so on. Thanks for your comments, we have added the references that you mentioned17. Parale, V. G.; Kim, T.; Choi, H.; Phadtare, V. D.; Dhavale, R. P.; Kanamori, K.; Park, H.-H. Mechanically Strengthened Aerogels through Multiscale, Multicompositional, and Multidimensional Approaches: A Review. Adv. Mater. 2024, 36(18), e2307772. https://doi.org/10.1002/adma.202307772
18. Tafreshi, O. A.; Mosanenzadeh, S. G.; Karamikamkar, S.; Saadatnia, Z.; Park, C. B.; Naguib, H. E. A Review on Multifunctional Aerogel Fibers: Processing, Fabrication, Functionalization, and Applications. Mater. Today Chem. 2022, 23, 100736. https://doi.org/10.1016/j.mtchem.2022.100736
Reviewer 2 Report
Comments and Suggestions for Authors
Author should cite more recent references of reviews or articles of aerogels.
Advanced Materials 36 (18) (2024) 2307772, Mater. Today Chem. 23 (2022) 100736, and so on.
Author Response
Reviewer 1: The comments have been addressed properly.Thanks for your comments